# Co-Designed Cardiac Rehabilitation for the Secondary Prevention of Stroke (CARESS): A Pilot Program Evaluation

**DOI:** 10.3390/healthcare12070776

**Published:** 2024-04-03

**Authors:** Sabah Rehman, Seamus Barker, Kim Jose, Michele Callisaya, Helen Castley, Martin G. Schultz, Myles N. Moore, Dawn B. Simpson, Gregory M. Peterson, Seana Gall

**Affiliations:** 1Menzies Institute for Medical Research, University of Tasmania, Hobart, TAS 7000, Australia; sabah.rehman@utas.edu.au (S.R.); seamus.barker@utas.edu.au (S.B.); kim.jose@utas.edu.au (K.J.); michele.callisaya@utas.edu.au (M.C.); myles.moore@utas.edu.au (M.N.M.); 2Peninsula Clinical School, Monash University, Frankston, VIC 3199, Australia; 3Royal Hobart Hospital, Tasmanian Health Service, Hobart, TAS 7000, Australia; helen.castley@ths.tas.gov.au; 4College of Health, Medicine and Wellbeing, University of Newcastle, Callaghan, NSW 2308, Australia; dawn.simpson@newcastle.edu.au; 5School of Pharmacy and Pharmacology, University of Tasmania, Hobart, TAS 7005, Australia; 6Faculty of Medicine, Dentistry and Health Sciences, Monash University, Melbourne, VIC 3800, Australia

**Keywords:** stroke, co-design, secondary prevention

## Abstract

Structured health system-based programs, such as cardiac rehabilitation, may reduce the risk of recurrent stroke. This study aimed to co-design and evaluate a structured program of rehabilitation, developed based on insights from focus groups involving stroke survivors and health professionals. Conducted in Tasmania, Australia in 2019, the 7-week program comprised one hour of group exercise and one hour of education each week. Functional capacity (6 min walk test), fatigue, symptoms of depression (Patient Health Questionnaire), and lifestyle were assessed pre- and post-program, with a historical control group for comparison. Propensity score matching determined the average treatment effect (ATE) of the program. Key themes from the co-design focus groups included the need for coordinated care, improved psychosocial management, and including carers and peers in programs. Of the 23 people approached, 10 participants (70% men, mean age 67.4 ± 8.6 years) completed the program without adverse events. ATE analysis revealed improvements in functional capacity (139 m, 95% CI 44, 234) and fatigue (−5 units, 95% CI −9, −1), with a small improvement in symptoms of depression (−0.8 units, 95% CI −1.8, 0.2) compared to controls. The co-designed program demonstrated feasibility, acceptability, and positive outcomes, suggesting its potential to support stroke survivors.

## 1. Introduction

It is estimated that there are 29,000 strokes each year in Australia [1], with around 400,000 people in the community who have had a stroke [2]. At least 80% of stroke episodes are caused by modifiable risk factors [3]. Therefore, the control of risk factors in people who have had a stroke is important and is recognised in clinical guidelines for the management of stroke. For example, the Australian Stroke Foundation’s Clinical Guidelines for the Management of Stroke recommend that “secondary prevention strategies should be considered for all patients with stroke or transient ischaemic attack (TIA) who are not receiving palliative care” [4]. Yet, the control of risk factors following stroke is disappointingly poor, including low smoking cessation [5], uncontrolled hypertension [6], and physical inactivity [7]. There is clear evidence for managing most stroke risk factors, which may include surgical and pharmacological interventions, along with lifestyle modifications. A 2023 overview of 15 systematic reviews of multimodal lifestyle-based interventions in people with stroke found moderate certainty evidence for lifestyle-based intervention, e.g., evidence they increased physical activity, and low certainty evidence for improved healthy eating and medication adherence [8]. Despite this evidence, at least 70% of stroke survivors in Australia report that they have unmet health needs, particularly in relation to secondary prevention [9], with 30% of people not receiving a formal discharge plan that included information and strategies to reduce their risk of another stroke [10].

One intervention that may improve secondary prevention of stroke is cardiac rehabilitation programs. These multimodal programs include the prescription of aerobic and resistance exercise and education to facilitate risk factor control and improved quality of life [11]. These programs are for people who have experienced or are at high risk of experiencing a cardiac event, thus the name refers to the patient group rather than the goal of improving cardiac function directly. In Australia, it is recommended that all cardiac patients be referred to these programs. Consequently, there are over 370 rehabilitation programs around Australia [12]. At face value, this type of program would appear to also address the needs of people with stroke in terms of risk factor management. However, in Australia [13] and other parts of the world [14], most cardiac rehabilitation programs do not include people with stroke, or they make up only a small percentage of users. The reasons for this are reported to include a lack of referrals, the complexity of including people with neurological deficits, potential safety risks (e.g., falls), and lack of resources [13,14,15]. Nonetheless, as a widely available, effective, embedded health system program, the potential adaptation and scale-up of cardiac rehabilitation programs to include stroke survivors is appealing.

Utilising cardiac rehabilitation programs to support secondary stroke prevention can be understood as a complex intervention [16] involving multiple interacting components that can be highly sensitive to the context of use. Changing the context from cardiac rehabilitation to a program for people with stroke requires an exploration of whether and how such a program should be adapted. Accordingly, the first aim of this project was to describe how to adapt cardiac rehabilitation as an intervention for secondary stroke prevention using co-design approaches among people who have had a stroke and health professionals. We hypothesized that it would be possible to adapt cardiac rehabilitation for people who have had a stroke using co-design processes. The second aim was to assess the feasibility of running the program through a pilot of the intervention, including evaluation of acceptability and association with health-related outcomes after stroke. We hypothesized that the program would be feasible and acceptable when pilot-tested within one health service.

## 2. Methods

The two aims of this project used distinct methods, which are outlined sequentially here: co-design, followed by delivery and evaluation of the pilot program. 

### 2.1. Co-Design

#### 2.1.1. Setting and Study Design

The aim of this study was to modify an existing cardiac rehabilitation program model for people with stroke at the Royal Hobart Hospital in Tasmania, Australia using the Promoting Action on Research Implementation in Health Services (PARIHS) framework in semi-structured focus groups to co-design the program [17]. 

#### 2.1.2. Participants

Focus groups were conducted in June 2018 with people with lived experience of stroke and health professionals, separately. We aimed to have 8 to 10 participants per focus group, based on published recommendations [18], and the experienced qualitative researcher who facilitated the discussions. This size was deemed suitable to ensure diversity of opinion but also a manageable discussion between members. Participants with lived experience were recruited from the Stroke Foundation (an advocacy organisation) and the local Stroke Support Group. Health professionals from the local health network who provided care to people with stroke in the primary, acute, or rehabilitation settings were recruited for the additional focus group. People who coordinated an existing cardiac rehabilitation program in the health service were also included in this focus group. This study was approved by the Tasmanian Human Research Ethics Committee (H0017243), and all the participants provided written informed consent.

#### 2.1.3. Process

The focus groups were held at the University of Tasmania. Background information about the management and secondary prevention of stroke, as well as an outline of existing cardiac rehabilitation programs, was provided. Workshops were facilitated by a senior qualitative researcher with experience leading focus groups. Discussion questions were prepared for each workshop by the research team and tailored to the perspectives of each group (Appendix A). Workshop discussions were audio-recorded, transcribed verbatim, and de-identified. Notes were taken during the workshops.

#### 2.1.4. Analysis

All the data were imported into NVivo (Lumivero, Denver, CO, USA) before being read and coded by an experienced qualitative researcher (KJ). Inductive coding identified and categorised codes before identifying key themes via a thematic analysis [19].

### 2.2. Pilot Program Development, Delivery, and Evaluation

#### 2.2.1. Program Development

Results of the thematic analysis, the existing local cardiac rehabilitation program, and clinical guidelines were used to design a 7-week, 2 h per week (1 h exercise, 1 h discussion) program (Figure 1). The program was delivered by a nurse, physiotherapist, and exercise physiologist, using existing resources developed by the Stroke Foundation for the education sessions [20].

#### 2.2.2. Program Delivery

##### Participants

The inclusion criteria were diagnosis of stroke, completed in-patient rehabilitation or discharged directly home from acute care at the Royal Hobart Hospital, and living at home. The Royal Hobart Hospital has the only stroke unit in the south of the state (population 250,000), with the statewide stroke protocol requiring all cases of stroke to be managed in that hospital. The exclusion criteria were moderate or severe dementia, and medically unstable or did not pass exercise screening (see below). We aimed for a sample size of 10 with two groups of 5 participants. This was for largely pragmatic reasons including the financial resources for the project and the size of the gym space available to conduct the group exercise. 

##### Recruitment

Potential participants from the hospital were approached by research or clinical staff and provided a brief information sheet (see online supplement). The support persons of the individuals with stroke were also encouraged to consent and participate in the program. If interested in participating, participants underwent a pre-exercise screening assessment with a stroke neurologist [21]. If deemed only suitable for “sedentary” activity, the participant was excluded. Study staff then conducted cognitive screening using the Telephone Interview for Cognitive Status (TICS) [22], with those scoring <19 excluded. The pilot study was approved by the Tasmanian Human Research Ethics Committee (H0017731 and H27593). The participants provided written informed consent before participation.

##### Pre-Program and Follow-Up Assessments

The participants completed a detailed health assessment with the study nurse before the program, and then 1 and 6 weeks after the program. The assessment involved baseline measurement of lifestyle (diet, physical activity, and smoking), stroke knowledge, health-related quality of life (AQoL8D), psychological well-being (Patient Health Questionnaire), fatigue, and biomedical risk factors (e.g., functional capacity, blood pressure, weight, lipids—see Appendix A). If the participant elected to have a support person attend the education component of the program, the support person completed an abridged pre-program assessment. At 1 and 6 weeks, additional data were collected on satisfaction and the self-reported impact of the program. All the data were collected electronically in REDCap. 

##### 7-Week Program: Exercise Component

Individual exercise programs were planned by an exercise physiologist, physiotherapist, and, if necessary, a neurologist. Sessions comprised a 15 min warm-up, a 30 min low- to moderate-intensity aerobic and strength training exercise circuit, and a 15 min cool down. Full details are provided in the supplement (Appendix A). Heart rate and self-reported exertion were monitored during the exercise sessions [23]. Any adverse events during the exercise sessions were recorded using a standard form.

##### 7-Week Program: Education Component

The study nurse led the education sessions (see Appendix A) using existing Stroke Foundation resources, which were adapted to the needs of the attendees. Open questions were used to facilitate discussion between group members. The sessions also encouraged the use of the Stroke Foundation’s EnableMe website to identify and track goals for individual attendees. 

#### 2.2.3. Evaluation of Program Feasibility, Acceptability, and Effectiveness

We evaluated the program in terms of feasibility and acceptability, as well as a preliminary examination of the effectiveness of the program on health-related outcomes, acknowledging the small sample size of this pilot study. 

Feasibility was assessed by the numbers screened, excluded, beginning, and finishing the program. Acceptability was examined using a descriptive analysis of survey questions related to the program at 1 week after the program concluded. 

Evaluation of the potential effectiveness of the program on outcomes was examined using paired Student’s *t*-tests or Fisher’s exact tests to compare means or proportions between time points before and 6 weeks after the program. We also used a historical control group from an observational study of physical activity after stroke in 2015–16 in the same population to undertake exploratory analyses of program effectiveness [24]. Common outcomes between the studies were the 6 min walk test and the fatigue assessment scale. As the measure of symptoms of depression in the historical study differed from our study, the scores were converted to z scores for analysis. We estimated the average treatment effect (ATE) and average treatment effect on the treated (ATET) by propensity score matching (1:2) with age and sex using programs ‘teffects’ and ‘psmatch’. Difference analyses were also used to estimate the effect of program participation by comparing the change in mean from baseline to follow-up between groups with propensity score matching. Complete case analysis was used. The analysis was performed using STATA 17 (StataCorp LLC, College Station, TX, USA).

## 3. Results

### 3.1. Program Co-Design

Nine participants attended the lived experience workshop (five people who were living with stroke, three spouses, and one person who had attended the local cardiac rehabilitation program), and eight people attended the health professional workshop (a pharmacist, a general practitioner, an aged care nurse, a neurologist, an exercise physiologist, a cardiac rehabilitation nurse, an occupational therapist, and a representative from the Heart Foundation). 

The data produced through the two focus groups were combined for purposes of analysis. From the thematic analysis of the data, we identified two primary themes: ‘existing post-stroke care’ and ‘adapted cardiac rehabilitation program considerations’, along with subthemes, as described below.

#### 3.1.1. Theme One: Existing Post-Stroke Care

##### Sub-Theme One: Isolation and Limited Support following Stroke

People with lived experience of stroke or cardiac disease and their supporters reported feeling isolated following discharge from the hospital system, and that ongoing support was lacking.

Health professionals highlighted delays for routine medical specialist follow-ups of up to three months due to waiting lists. Survivors of stroke were routinely provided with an information pack in the hospital with information developed by the Stroke Foundation. However, stroke survivors found this information overwhelming, given the acuity of their condition, and would have appreciated it if someone had discussed the information pack with them. Health professionals preferred to tailor information to the individual needs of each stroke survivor but reported that it was difficult to predict those needs during the period of hospitalisation. 

##### Sub-Theme Two: Risk Factor Management

Survivors of stroke could not recall any specific discussions about managing risk factors that increased the risk of a subsequent stroke. Health professionals speculated that people with mild stroke or those who recovered fully might “trivialise their stroke” and return to previous lifestyle habits. Health professionals expressed concern that, after hospital discharge, consistency in the medical management of risk factors, such as high blood pressure, was made more difficult due to breakdowns in communication across the continuum of care. This continuum was complex for many people, involving transitions from the acute hospital to inpatient rehabilitation, and then to primary care in the community (under the auspices of a general practitioner).

#### 3.1.2. Theme Two: Adapted CR Program Considerations

##### Sub-Theme One: Benefits

People with lived experience were overwhelmingly supportive of a program they could attend following stroke. The primary benefit they identified was meeting peers who had experienced stroke, as well as ameliorating isolation experienced after stroke. Health professionals recognised that “a support network is invaluable in the early recovery phase”, although some were surprised that peer support was such a strong focus for people following a stroke. Survivors of stroke and their caregivers valued a point of human contact in a program who could answer any questions. 

##### Sub-Theme Two: Processes

The processes associated with cardiac rehabilitation, such as referral pathways, which would need to be adapted to suit people with stroke, were discussed. Concerns were raised regarding not overburdening stroke survivors too soon after their stroke, as well as not conflicting with a process of stroke rehabilitation taking place in hospital or community settings. Participants suggested that the timing of referral to the program would need to be carefully considered to optimally fit within the overall continuum of care of the stroke survivor. One suggestion, by a health professional, was for referral to take place after discharge from the stroke rehabilitation process. Another suggested that a referral mechanism was needed that accounted for the vast number of points within the continuum of care from which a referral to the program could potentially come. The need to promote the program to referrers, as well as stroke survivors, was emphasised.

Another process that was discussed was that of assessment and the need for appropriate program inclusion and exclusion criteria. Additionally, it was suggested that the prescription of a tailored, rather than generic, exercise program would require an assessment of the specific physical issues of the individual stroke survivor by an appropriate allied health professional. 

##### Sub-Theme Three: Format

People with lived experience of stroke were emphatic that any group needed to be “very informal” and focused on life experiences and peer-to-peer learning, rather than didactic in style. People with lived experience of stroke suggested that having written materials to take home with them would assist in retaining information, acknowledging that their cognitive function and memory could be compromised following stroke. Educational material would ideally be developed by a multidisciplinary team, who may not need to be involved after the development phase. The inclusion of *carers* in the education component was acknowledged as important by health professionals, survivors of stroke, and the carers themselves, who were also seeking sources of peer support. In terms of length, it was suggested that six weeks (the length of the current cardiac rehabilitation program at the hospital) would be too short to effect behaviour change, and so a longer program, potentially including intermittent follow-up, was favoured. It was suggested that nurses could be program facilitators but with professional input for the exercise component from physiotherapists or exercise physiologists. In terms of participants, people with lived experience of stroke advocated that stroke survivors with mobility limitations could attend the program. Health professionals suggested that clustering participants into cohorts, based on shared functional impacts, or age, could allow a “targeted approach to the program”, but there was no consensus reached on this point. People with lived experience suggested that the program could serve as a common point of case coordination and provide referrals, as needed, to other services. 

##### Sub-Theme Four: Content

People with lived experience of stroke or cardiac disease felt that the inclusion of exercise in the rehabilitation program was “a terrific idea”. Health professionals, however, raised concerns regarding the variability of physical function following stroke, and about exercise prescription. One concern was not wanting to provide exercises that were different to those being prescribed by the providers of stroke rehabilitation, due to the potential to confuse the patient or work at cross-purposes. One health professional felt that “targeted therapy” should be the purview of stroke rehabilitation, not a modified cardiac rehabilitation program. In contrast, another health professional felt stroke survivors should be prescribed an individualised program, despite the challenges of delivering that with supervision in a group environment. This issue also prompted the suggestion that exercise-focused allied health professionals would need to be involved in the program. 

People with lived experience of stroke and health professionals identified the importance of addressing the impact of stroke on mental health and how this can, in turn, impact recovery. Stroke survivors suggested that this topic was not well covered during hospital admission despite its importance, and so should be included in a stroke-adapted cardiac rehabilitation program. People with lived experience of stroke also suggested that a modified cardiac rehabilitation program should emphasise that stroke is a “chronic disease” with effects that can last “the rest of your life”. 

A person who had previously completed the local cardiac rehabilitation program described how it covered “cardio health”, including the role of smoking, drinking, obesity, and genetics. These were considered suitable topics for the lifestyle education component by people with lived experience, although they preferred an informal, peer-to-peer format.

### 3.2. Program Delivery and Evaluation

Of the 23 people approached to participate in the study, 10 (43%) participated in two groups of 5 people (Figure 2). Only two participants invited a support person to participate in the program. Only one support person completed both the baseline and follow-up assessments, so the results are not presented here. The reasons for non-participation included returning to work, non-response to contact, and not being interested in participating. Completion of each week of the program varied between 100% and 60% across the two groups. The characteristics of the participants are shown in Table 1.

Satisfaction and impact were rated highly by participants at 1 week after program completion (n = 8, 80% completed assessment, Table 2). All the participants recommended the program to others and agreed/strongly agreed that the program helped them to learn about their health and encouraged them to take better care of themselves and make lifestyle changes.

Knowledge about stroke increased from baseline to follow-up at 1 or 6 weeks after program completion (Appendix A). For example, the proportion correctly reporting stroke risk factors increased from baseline to 1 and 6 weeks, including for high blood pressure (20% baseline, 88% 6-week follow-up); smoking (50% baseline; 63% 6-week follow-up); and overweight (40% baseline; 63% 6-week follow-up). Compared to before the program, a greater proportion of people also identified the FAST signs and stated they would call an ambulance if having stroke symptoms.

There was some evidence that the program was associated with positive improvements in health outcomes (Table 3). In analyses comparing baseline to 6 weeks after the program, some positive changes in quality of life, fatigue, symptoms of depression, functional capacity, and fruit and vegetable intake were observed, although these did not reach statistical significance. There was a slight increase in systolic and diastolic blood pressures but no change in total cholesterol, body mass index or grip strength from baseline to follow-up.

The program findings were also compared to a historical control group of 31 people from an observational study in the same region (see online Appendix A for characteristics). Propensity score matching with age and sex was performed to compare the outcomes of program participants (n = 7) with historical controls (n = 14). These analyses demonstrated that the program participants had improvements in functional capacity on the 6 min walk test, fatigue levels, and depression symptoms compared to the historical controls (Table 4).

## 4. Discussion

We successfully co-designed and piloted a modified cardiac rehabilitation program to enhance risk factor management after stroke. In accordance with the evidence base and the co-design process, the exercise completed in our program was tailored and delivered by allied health professionals who progressed the exercise over the program. It was designed to result in physiological changes that modify known risk factors. Similarly, the education and discussion in the program were designed to begin behavioural changes and improve psychological well-being. The program was feasible and acceptable in a non-randomised pilot study based in a single tertiary hospital. The exploratory analyses showed potentially positive changes in health outcomes 6 weeks after the program when compared to a historical control group. 

Compared with pre-program participation, small but positive associations with improved quality of life, fatigue, symptoms of depression, functional capacity, and stroke knowledge were found. Importantly, we were able to use a historical control group to show that changes in fatigue, depression symptoms, and functional capacity were greater than those seen in the usual recovery phase without program participation [24]. Our findings, albeit in a small sample from one hospital network, are supported by a growing body of literature demonstrating the benefits of similar, multimodal programs on a range of cardiovascular risk factors [25,26,27,28,29]. Recent cohort studies provide preliminary evidence that a modified cardiac rehabilitation program can decrease mortality, hospital readmissions, and subsequent strokes among stroke survivors [30,31]. Participation in these types of programs has been shown to increase participants’ self-efficacy, coping and resilience, albeit in cardiac populations [32,33], which potentially has a compounding effect on both cardiovascular health and general well-being, resulting in greater outcomes than the sum of the individual components. 

An important feature of the program was the use of co-design, including people with lived experience of stroke and health professionals. Co-design assisted with practical aspects of program design and delivery and provided insights into the potential wider benefits of such a program. Among people with lived experience of stroke, a strong rationale for this type of group program was that it could ameliorate isolation, provide a source of peer support, and be a centralised point of contact for the stroke survivor. The co-design process was confirmatory in terms of discussion highlighting the fragmented care people with stroke were receiving when returning to the community, the lack of guidance on risk factors and the importance of other aspects of recovery, including mental health, which has also been found in quantitative studies [9,10]. 

Given the strong rationale for this type of program, including existing evidence of effectiveness, the next steps could be a hybrid implementation and effectiveness trial [34]. The very high rating of program satisfaction, attendance, and perceived impact support this approach. One reason for us proposing this type of program and working together with the existing team who deliver a similar program to people with cardiac disease was that, theoretically, implementation should be more straightforward. However, as noted by others, many barriers exist to the implementation of new programs as well as the integration of people with stroke into existing programs [15,35]. Among these barriers, funding for such a program is particularly problematic. While all hospital networks with stroke units fund comprehensive rehabilitation services focused on recovery of function, we are aware of very few that fund this type of program focused on more holistic aspects of ‘recovery’. It appears inequitable to have hundreds of state-funded services across the country seemingly directed at all cardiac patients while, for people with stroke, it is only those with physical limitations—approximately 50%—who can access state-funded care through rehabilitation services. Difficulties related to the division of care between hospitals, which are state-funded, and primary or community care, which is federally funded, likely contribute to this issue [36]. 

The limitations of the pilot study include the small number of participants from a single centre. This was overcome to some extent by using the historical control group to compare outcomes over time. The characteristics of the sample show they were younger, more often male and with worse co-morbidities than other similar post-stroke populations [37]. The study was based at one hospital site, limiting external validity; however, it is the only hospital with a stroke unit in the region, meaning the patients who attend that hospital are representative of the broader community. The co-design process included people with lived experience and health professionals guided by an experienced facilitator; however, the groups were small and, if repeated with further groups, may have yielded different results. The study was conducted prior to the COVID-19 pandemic. Care for people with stroke, and other conditions, within the hospital system has changed considerably since then, with face-to-face and group programs suspended [38]. We did not have a comprehensive assessment of all important outcomes for people who have had a stroke. For example, we assessed symptoms of depression but not other psychological factors such as stress, anxiety, or coping. While we intended to recruit support people, only one such person participated, so we could not do any meaningful analyses. However, the inclusion of support people was noted to be important during the co-design, so future studies or programs should include this group to potentially assist with the high burden they experience [39]. The main strengths of the study were the use of co-design and the leveraging of an existing model of health care within the hospital system.

## 5. Conclusions

We found that a co-designed program in one hospital for people after stroke modelled on cardiac rehabilitation was feasible, acceptable, and associated with small but positive changes in outcomes important for people after stroke. Future studies should focus on expanding the evidence base for the effectiveness and implementation of such programs.

## Figures and Tables

**Figure 1 healthcare-12-00776-f001:**
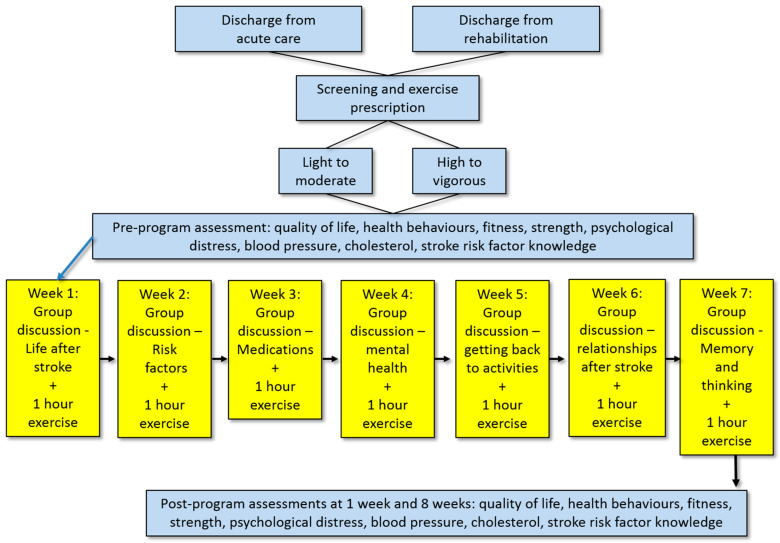
Co-designed CARESS program including screening, assessments, and content.

**Figure 2 healthcare-12-00776-f002:**
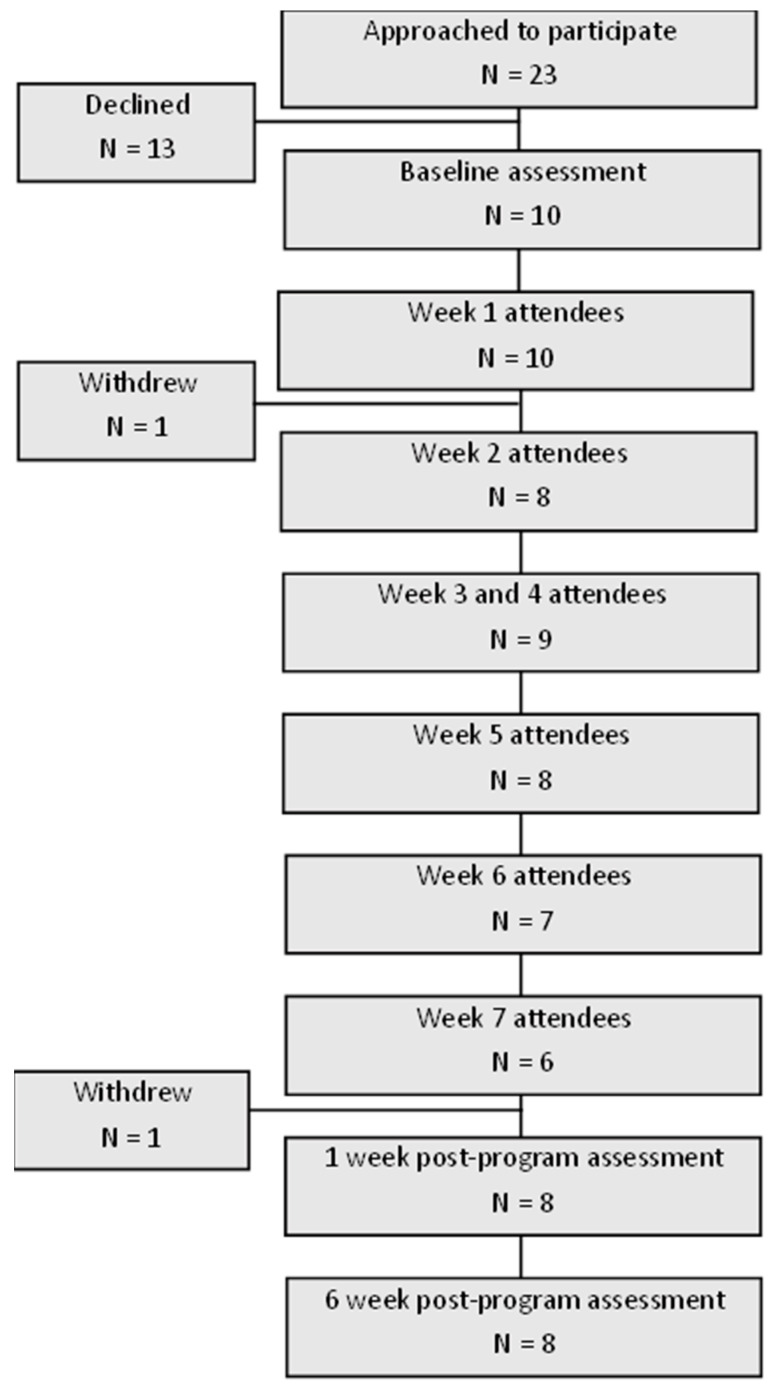
Flow chart of participation in CARESS pilot program.

**Table 1 healthcare-12-00776-t001:** Characteristics of participants at baseline (n = 10) in the CARESS program.

	n/Mean	%/sd
Age (years) mean (SD)	66.5	9.12
Sex		
Men	7	70
Women	3	30
Type of stroke		
Ischaemic	10	100
Haemorrhagic	-	
Time from stroke (weeks)	12.47	6.98
BMI	29.8	5.63
Lifetime smoking > 100 cigarettes		
No	3	30
Yes	7	70
Current smoker		
No	10	100
Yes	0	0
History of hypertension		
No	1	10
Yes	9	90
Hypercholesterolaemia		
No	3	30
Yes	7	70
Diabetes		
No	9	90
Yes	1	10
Atrial fibrillation		
No	9	90
Yes	1	10

**Table 2 healthcare-12-00776-t002:** Satisfaction and impact of CARESS program reported 1 week after completion.

Satisfaction with Program	N	Mean (SD)/%
How would you rate this program? (1 = worst possible program, 10 = best possible program)	8	8.6 (0.77)
Would you recommend this program to other people who have had a stroke?		
Yes	8	100%
No	0	0%
Overall, how satisfied were you with the program? (1= very satisfied, 5 = very unsatisfied)	8	1.12 (0.35)
Impact of program		
The program helped me to understand my health issues (1 = strongly agree, 5 = strongly disagree)	8	1.37 (0.51)
Strongly agree/agree	8	100%
Disagree/strongly disagree	0	0
The program helped me to learn ways to take better care of myself (1 = strongly agree, 5 = strongly disagree)	8	1.25 (0.46)
Strongly agree/agree	8	100%
Disagree/strongly disagree	0	0

**Table 3 healthcare-12-00776-t003:** Comparison of functional capacity, fatigue, and symptoms of depression between CARESS program participants (n = 7) and historical controls (n = 14) with propensity score matching on age and sex.

	Average Treatment Effects Analysis		Difference-in-Differences (DID) Analysis				
			Program Participants		Historical Controls		
	ATE (95% CI)	ATET (95% CI)	Baseline Mean (95% CI)	Follow-Up Difference Mean (95% CI)	Baseline Mean (95% CI)	Follow-Up Difference Mean (95% CI)	DiD (95% CI)
6 min walk test (meters)	139 (44, 234) *	117 (−25, 262)	350 (153, 547)	117 (−69, 304)	337.3 (209, 466)	66 (−10, 143)	51 (−215, 317)
Fatigue	−5 (−9, −1) *	−4 (−9, 2)	22 (13, 30)	−4 (−9, 2)	19 (13, 25)	2 (−4, 7)	−5 (16, 5)
Symptoms of depression z-scores	−0.8 (−1.8, 0.2)	−0.5 (−1.3, 0.4)	−0.1 (−1.1, 1.1)	−0.7 (−1.3, −1.5)	−0.4 (−0.7, −0.1)	0.1 (−0.4, 0.8)	−0.9 (−2.2, 0.5)

* *p* < 0.05.

**Table 4 healthcare-12-00776-t004:** Health outcomes before and 6 weeks after the CARESS program.

	Baseline		6-Week Follow-Up		
Outcome Measure	n/Mean	%/sd	n/Mean	%/sd	*p*-Value
Quality of Life (AQoL) utility score	0.74	0.21	0.84	0.12	0.25
Balance test mean (SD)	24.7	3.40	22.3	5.18	0.26
Symptoms of depression (z-score) mean (SD)	0.11	1.1	−0.65	1.1	0.19
Fatigue score mean (SD)	22.1	8.38	18.87	4.51	0.34
6 min walk test (meters) mean (SD)	397.67	169.68	490.12	124.01	0.22
Fruit and vegetable intake per day					0.18
0–1 servings	0	0	0	0	
2–3 servings	10	100	6	75	
4 or more servings	0	0	2	25	
Physical activity MET (min/week)	3326.5	4692.77	3174.85	3562.45	0.94
Body mass index	30.86	6.36	30.79	6.03	0.88
Grip strength					
Right	31.42	13.00	32.22	12.55	0.55
Left	26.39	11.44	26.46	13.86	0.97
Total cholesterol (mmol/L)	3.1	0.44	3.1	0.82	0.88
Systolic blood pressure (mmHg)	128.4	15.1	138.8	18.0	0.21
Diastolic blood pressure (mmHg)	78.8	15.5	83.7	12.4	0.48

## Data Availability

The data presented in this study are available on request from the corresponding author.

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
