# Peer review of "Co-Designed Cardiac Rehabilitation for the Secondary Prevention of Stroke (CARESS): A Pilot Program Evaluation"

_healthcare, 2024, doi:10.3390/healthcare12070776_

Round 1

Reviewer 1 Report

Comments and Suggestions for Authors

Hello, dear colleagues!

Your study was compiled according to all the rules and the relevance is quite high, taking into account the epidemic of the disease, socio-economic impact and the general literacy of the medical community.

It is necessary to note the high percentage of originality of the text and the quality of presentation.

I have almost no questions, but I’ll ask anyway

1. What was the purpose of the study? this is not highlighted in the text

2. What is your null hypothesis?

3. How did you calculate the sample size and where in the text do you say this?

4. The conclusion section is missing or not highlighted with a subheading

5. The list of references should be updated due to the possible expansion of the introduction, where it is necessary to focus on the prevalence of the disease in the age-related population

Author Response

Dear Reviewer 1,

Reviewer comments in bold, author response in plain text and revised text in italics.

Reviewer 1

  1. What was the purpose of the study? this is not highlighted in the text

The aim of the study was given in the introduction:

Line 70-72: Accordingly, the first aim of this project was to describe how to adapt cardiac rehabilitation as an intervention for secondary stroke prevention using co-design approaches among people who have had a stroke and health professionals. The second aim was to assess the feasibility of running the program through a pilot of the intervention, including evaluation of acceptability and association with health-related outcomes after stroke.

  1. What is your null hypothesis?

We have now added hypotheses to the introduction after each aim:

Line 73-74: We hypothesized that it would be possible to adapt cardiac rehabilitation for people that have had a stroke using co-design processes.

Line 76-77: We hypothesized that the program would be feasible and acceptable when pilot tested within one health service.

  1. How did you calculate the sample size and where in the text do you say this?

We have now added information about the sample size for the focus groups and pilot study.

Focus groups (lines 90-93):

We aimed to have 8 to 10 participants per focus group based on published recommendations [18] and the experienced qualitative researcher that facilitated the discussions. This size was deemed suitable to ensure diversity of opinion but also manageable discussion between members.

Pilot study (lines 130-133)

We aimed for a sample size of 10 with two groups of 5 participants. This for largely pragmatic reasons including the financial resources for the project and the size of the gym space available to conduct the group exercise.

  1. The conclusion section is missing or not highlighted with a subheading

We have added the conclusion subheading on line 428

  1. The list of references should be updated due to the possible expansion of the introduction, where it is necessary to focus on the prevalence of the disease in the age-related population

Thank you for your suggestion. We have added an opening sentence that provides information on the incidence and prevalence of stroke in Australia. We have subsequently added two references from the Australian Institute of Health and Welfare.

Line 32: It is estimated that there are 29,000 strokes each year in Australia [1] with around 400,000 people in the community that have had a stroke.[2]

  1. Australian Institute of Health and Welfare. Estimating the incidence of stroke and acute coronary syndrome using the National Integrated Health Services Information Analysis Asset, catalogue number CDK 21. Canberra: 2022.
  2. Australian Institute of Health and Welfare. Heart, Stroke and vascular disease: Australian facts. Canberra: 2023.

Reviewer 2 Report

Comments and Suggestions for Authors

In the current manuscript “Cardiac Rehabilitation for the secondary prevention of Stroke (CARESS): Co-design and evaluation of a pilot program to support people after Stroke” the authors propose a structured program with a cardiovascular health centred focus, to aid stroke survivors in rehabilitation and risk reduction for reoccurring stroke.

Although this is a relatively feasible rehabilitation program, focussing on a very important pathology that required extensive rehab programs, to ensure the validity and power of this program, a few major concerns should be addressed:

·        As this is a cardiovascular/ cardiac focussed rehabilitation program, most of the interventions are exercise related – thus more cardiac-respiratory fitness focus, than cardiac.

·        A very important aspect that is missing is the time from stroke – when did these patients suffer a stroke (5 months ago, a year ago etc) – this is an extremely important aspect to take into account when, especially evaluating factors such as depressive symptoms, mobility, perception of success. This is a key aspect to report given the small sample size and possible skewness of distribution.

·        Most of the data is self-report, however to evaluate the validity and success of such an intervention from a clinical perspective, data on BP, HR, lipids and glucose should be included (data gathered at baseline versus last contact session of this pilot study)

·        The PHQ is used as a marker of depressive symptoms, and only values exceeding 10 can be considered medium to high/ severe depressive symptoms – this is not a diagnostic toll for depression, thus please rephrase.

·        As the PHQ is used to evaluate depressive symptoms, it is important to note, especially for ischemic stroke (which is the subtype all 10 participants had), that stress (stress, resilience and coping) are some of the main factors not only involved in stroke etiology but also the main psychophysiological factors that have to be considered in stroke intervention studies (please see the following review) https://academic.oup.com/abm/article/57/2/111/6605937 . Additionally, as this was a co-designed program (including health care workers), please also note the significant effect of physiological strain on care givers to those post stroke (https://www.ncbi.nlm.nih.gov/pmc/articles/PMC10172376/ )

Indeed, although this is an intervention program with a more CV focus, physiological stress, coping and resilience show significant effects on the CV system (Sympathetic hyperactivity, vagal withdrawal, increased cardiac strain, risk for arrythmias etc). The authors are urged to consider these aspects, especially given the stroke etiology.

·        Detail regarding the informed consent procedure should please be provided.

Author Response

Dear Reviewer 2,

Reviewer comments in bold, author response in plain text and revised text in italics.

Reviewer 2

  1. As this is a cardiovascular/ cardiac focussed rehabilitation program, most of the interventions are exercise related – thus more cardiac-respiratory fitness focus, than cardiac.

Yes, we agree that the exercise intervention is primarily focused on improving cardiorespiratory and muscular fitness rather than cardiac functions specifically. The term ‘cardiac rehabilitation’ is the common term for these programs in Australia. We have added some information to clarify this in the introduction:

Line 53-55: These programs are for people that have experienced, or are at high risk of experiencing, a cardiac event, thus the name refers to the patient group rather than a goal of improving cardiac function directly.

  1. A very important aspect that is missing is the time from stroke – when did these patients suffer a stroke (5 months ago, a year ago etc) – this is an extremely important aspect to take into account when, especially evaluating factors such as depressive symptoms, mobility, perception of success. This is a key aspect to report given the small sample size and possible skewness of distribution.

      Thank you for alerting us to the omission of this information. We have now added this to Table 1 and the supplementary table 6.

  1. Most of the data is self-report, however to evaluate the validity and success of such an intervention from a clinical perspective, data on BP, HR, lipids and glucose should be included (data gathered at baseline versus last contact session of this pilot study)

      We did not record data on heart rate or glucose. We did have objectively measured data on blood pressure, cholesterol, and functional capacity, as reported in Table 3. In addition, we had measured data on BMI and grip strength that we had not reported, so have added those measures to Table 3. We have added text to the results section describing differences between baseline and follow-up.

Lines 337-339: There was a slight increase in systolic and diastolic blood pressures but no change in total cholesterol, body mass index or grip strength from baseline to follow-up.

  1. The PHQ is used as a marker of depressive symptoms, and only values exceeding 10 can be considered medium to high/ severe depressive symptoms – this is not a diagnostic toll for depression, thus please rephrase.

      We agree that this is an important point. We have now edited the text to ensure that rather than saying ‘depression’ we have stated ‘symptoms of depression’ or ‘depression symptoms’.

See edits on lines 19, 26, 187, 314, Table 3, 335, Table 4, 345.

  1. As the PHQ is used to evaluate depressive symptoms, it is important to note, especially for ischemic stroke (which is the subtype all 10 participants had), that stress (stress, resilience and coping) are some of the main factors not only involved in stroke etiology but also the main psychophysiological factors that have to be considered in stroke intervention studies (please see the following review) https://academic.oup.com/abm/article/57/2/111/6605937 .

It is a limitation that the measure used only captured depression symptoms. We have added this to the discussion.

Line 419-421: We did not have a comprehensive assessment of all important outcomes to people that have had a stroke. For example, we assessed symptoms of depression but not other psychological factors such as stress, anxiety or coping.

  1. Additionally, as this was a co-designed program (including health care workers), please also note the significant effect of physiological strain on care givers to those post stroke (https://www.ncbi.nlm.nih.gov/pmc/articles/PMC10172376/ )

This is an important point. We did attempt to include caregivers but only had one participate. We have added some information to the limitation sections about this issue.

Line 421-425: While we intended to recruit support people, only one such person participated so we could not do any meaningful analyses. However, inclusion of support people was noted to be important during co-design, so future studies or programs should include this group to potentially assist with the high burden they experience.[37]

  1. Indeed, although this is an intervention program with a more CV focus, physiological stress, coping and resilience show significant effects on the CV system (Sympathetic hyperactivity, vagal withdrawal, increased cardiac strain, risk for arrythmias etc). The authors are urged to consider these aspects, especially given the stroke etiology.

The non-exercise components of the program may have influenced these factors that could, in turn, affect cardiovascular function, as you suggest. To address this issue, we have added some further detail to the discussion:

Lines 376-378: Participation in these types of programs have been shown to increase participants’ self-efficacy, coping and resilience, albeit in cardiac populations,[31, 32] which potentially has a compounding effect on both cardiovascular health and general wellbeing, resulting in greater outcomes than the sum of the individual components.

  1. Detail regarding the informed consent procedure should please be provided.

      We have amended the methods and relevant section at the end of the document to include the IRB details: lines 98-100, 143-144, 455-457.

Reviewer 3 Report

Comments and Suggestions for Authors

Title: " Cardiac Rehabilitation for the secondary prevention of stroke (CARESS): co-design and evaluation of a pilot program to support people after stroke".

Strengths of the Article:

This study aimed to modify an established cardiac rehabilitation model for individuals recovering from stroke at the Royal Hobart Hospital in Tasmania, Australia. Employing the Promoting Action on Research Implementation in Health Services (PARIHS) framework, the adaptation process engaged semi-structured focus groups with both stroke survivors and healthcare professionals. The primary objectives were twofold: firstly, to elucidate the transformation process of cardiac rehabilitation into a secondary stroke prevention intervention through collaborative design with stroke survivors and healthcare professionals. Secondly, the study assessed the pilot program's feasibility, encompassing its acceptability and correlation with health-related outcomes post-stroke. 

The program's Average Treatment Effect (ATE) was determined using propensity score matching. Key themes identified from the co-design focus groups emphasized the importance of coordinated care, improved psychosocial management, and the inclusion of carers and peers in program development. Out of the 23 participants, only 10 completed the program without any adverse events. ATE analysis revealed significant enhancements in functional capacity and fatigue, with a modest reduction in depression compared to the control group. The co-design process validated discussions around the fragmented care that stroke survivors experienced upon returning to the community, the lack of guidance on risk factors, and the importance of addressing various aspects of recovery, including mental health.

This co-designed program demonstrated feasibility, acceptability, and positive outcomes, suggesting its potential efficacy in supporting individuals recovering from stroke. The program exhibited significant improvements in functional capacity and fatigue, along with a modest reduction in depression compared to controls. The positive results underscore the benefits of incorporating stroke survivors in program development, addressing their specific needs for coordinated care, improved mental health support, and peer connection.

While the manuscript content is robust, some critical refinements are suggested to enhance clarity.

 Major Critiques and Suggestions:

1. Sample Size and Applicability Concerns: One significant concern revolves around the study's limited applicability due to the small sample size. Only 10 participants engaged in the study, further divided into groups of 5 individuals each (as depicted in Figure 2). Moreover, Table 4 compares only 7 participants against 14 historical controls. From my perspective, the study findings, based on such a diminutive sample, raise questions about their broader relevance to diverse groups of stroke survivors. The generalizability and robustness of the conclusions might be compromised given the narrow scope of participant representation.

2. Single-Center Recruitment Limitations: The limitations of the pilot study extend beyond the small participant number to the exclusive recruitment from a single center. Despite the authors' attempt to mitigate this by using a control group for temporal outcome comparisons, the inherent constraints persist.

-Relying solely on a single center introduces potential biases and restricts the study's external validity. This limitation hinders the extrapolation of findings to a more diverse and representative population of stroke survivors.

3. A dedicated section for limitations encourages readers to critically engage with the study's methodological challenges.

4. Table 4 Reference Error: Line-323; (Table 4Error! Reference source not found.)?

 5. Incomplete Statements:  Institutional Review Board Statement:

   Informed Consent Statement:

   Data Availability Statement:

   Why are these left blank?

Minor Points:

1. Suggested Title: "Co-Designed Cardiac Rehabilitation for Secondary Stroke Prevention (CARESS): A Pilot Program Evaluation."  I believe this title is more appropriate and focused.

2. The sequence of the tables presented needs to be changed, as Table 4 comes before Table 3.

3. Line 167 - 169; why is a different font size used?

4. Lines 171 & 274; why 'Bold' fonts were used? If it's a typo, correct this.

Comments on the Quality of English Language

The English language is fine some minor changes are required.

Author Response

Dear Reviewer 3,

Reviewer comments in bold, author response in plain text and revised text in italics.

Reviewer 3

 Major Critiques and Suggestions:

  1. Sample Size and Applicability Concerns: One significant concern revolves around the study's limited applicability due to the small sample size. Only 10 participants engaged in the study, further divided into groups of 5 individuals each (as depicted in Figure 2). Moreover, Table 4 compares only 7 participants against 14 historical controls. From my perspective, the study findings, based on such a diminutive sample, raise questions about their broader relevance to diverse groups of stroke survivors. The generalizability and robustness of the conclusions might be compromised given the narrow scope of participant representation.

We agree that the small size of the pilot study is a major limitation, which was the result of limited funding to run the pilot. We have added further detail about the sample size to the methods (lines 130-133). We also made some comparisons between our study sample and a similar stroke population in the discussion (lines 410-412). See also responses to next comment for relevant edits.

We aimed for a sample size of 10 with two groups of 5 participants. This was for largely pragmatic reasons including the financial resources for the project and the size of the gym space available to conduct the group exercise.

The characteristics of the sample show they were younger, more often male and with worse co-morbidities than other similar post-stroke populations.

  1. Single-Center Recruitment Limitations: The limitations of the pilot study extend beyond the small participant number to the exclusive recruitment from a single center. Despite the authors' attempt to mitigate this by using a control group for temporal outcome comparisons, the inherent constraints persist.

-Relying solely on a single center introduces potential biases and restricts the study's external validity. This limitation hinders the extrapolation of findings to a more diverse and representative population of stroke survivors.

We agree that this is a major limitation and have added more discussion regarding this point. We have also added further information to the methods about the hospital site as this is the only facility that has a stroke unit in the area that is required to manage all cases according to the statewide stroke protocol.

Line 127-129: The Royal Hobart Hospital has the only stroke unit in the south of state (population 250,000) with the statewide stroke protocol requiring all cases of stroke to be managed in that hospital.

Line 363: The program was feasible and acceptable in a non-randomised pilot study based in a single tertiary hospital.

Line 412-414: The study was based at one hospital site limiting external validity; however, it is the only hospital with a stroke unit in the region meaning the patients that attend that hospital are representative of the broader community. The co-design process included people with lived experience and health professionals guided.

Line 431: We found that a co-designed program in one hospital for people after stroke modelled on cardiac rehabilitation

  1. A dedicated section for limitations encourages readers to critically engage with the study's methodological challenges.

We have now separated the limitations section to make it clearer in the text (line 408)

  1. Table 4 Reference Error: Line-323; (Table 4Error! Reference source not found.)?

We have checked all table captions and cross-references to ensure there are not formatting errors.

  1. Incomplete Statements:  Institutional Review Board Statement:

   Informed Consent Statement:

   Data Availability Statement:

   Why are these left blank?

These have now been completed. See lines 455-461.

Minor Points:

  1. Suggested Title: "Co-Designed Cardiac Rehabilitation for Secondary Stroke Prevention (CARESS): A Pilot Program Evaluation."  I believe this title is more appropriate and focused.

Thank you for this suggestion. We have amended as suggested (line 2-3).

  1. The sequence of the tables presented needs to be changed, as Table 4 comes before Table 3.

Thank you for alerting us to this issue. We have changed the table numbers and checked that the text follows in the same sequence. Note that paragraph on page 11 was moved earlier (lines 340-345) to come before Table 4.

  1. Line 167 - 169; why is a different font size used?

We have checked formatting of fonts throughout the text to ensure they are consistent.

  1. Lines 171 & 274; why 'Bold' fonts were used? If it's a typo, correct this.

We have checked formatting of fonts throughout the text to ensure they are consistent.

Round 2

Reviewer 2 Report

Comments and Suggestions for Authors

All comments have been adequately addressed, thank you.

Comments on the Quality of English Language

N/A

Reviewer 3 Report

Comments and Suggestions for Authors

In the revised manuscript authors have successfully addressed the comments raised by the reviewer and incorporated suggestions to improve the quality of the paper. I think this manuscript has been improved from the previous version.